# Laminate Design of Carbon-Fiber-Reinforced Resin Matrix Composites for Optimized Mechanical Properties and Electrical Conductivity

**DOI:** 10.3390/ma15227876

**Published:** 2022-11-08

**Authors:** Hongxue Tan, Yanxiang Wang, Chengguo Wang, Chengjuan Wang, Mengfan Li, Haotian Jiang, Zhenhao Xu

**Affiliations:** 1Key Laboratory for Liquid−Solid Structural Evolution and Processing of Materials (Ministry of Education), Shandong University, Jinan 250061, China; 2Carbon Fiber Engineering Research Center, School of Materials Science and Engineering, Shandong University, Jinan 250061, China

**Keywords:** composite material, pantograph slide plate, carbon fiber, laminated structure

## Abstract

Carbon fiber composites as pantograph slide materials are in the development stage, in which copper is the conductive phase, and the addition form and size need to be designed. Herein, the effects of the copper morphology, the size of the copper mesh on the performance, and the influence of the contact mode between the sliding plate and bracket on the temperature rise were compared and analyzed. The resistivity is 11.2 μΩ·m with the addition of 20 wt% copper mesh, a relative reduction of 91.77%. Importantly, the impact strength is increased by 14.19%, and the wear is reduced by 13.21%; hence, the copper mesh laid in layers is the ideal structure. Further study of the distribution and quality of the copper mesh shows that the resistivity is related only to the quality of the copper mesh; in addition, the number of layers of the copper mesh cannot exceed 16, and it is determined that the best type of copper mesh is 5#. Notably, the performance can be improved by appropriately reducing the thickness of the copper mesh and increasing the aperture while the sliding plate and the bracket are connected by copper mesh with conductive adhesive, which has the slowest heating rate of 2.27 °C/min and the smallest resistance. Therefore, the influence of copper content and distribution on the electrical conductivity are systematically investigated, and the mechanical properties and electrical conductivity are optimized through the design of the laminate structure of the compound material.

## 1. Introduction

One of the important indicators of measuring national transportation is the degree of railway electrification. Electric locomotives which are not only economical, high-speed, and heavy-duty, but also have a series of advantages such as being energy-saving and pollution-free—play an important role in modern railway transportation [1,2,3,4]. With the rapid development of modern railway transportation, the research of pantograph slide plates as the key device of power transmission has been widely considered [5]. As the soul of the high-speed electric locomotive, the pantograph slide plate is the current-collecting element from which the electric locomotive obtains power from the conductor of OCS. The sliding plate is installed on the top of the locomotive [6] and directly contacts the contact conductor [7], which will flow the current of the 25 kV voltage contact network to provide power for the electric locomotive [8].

The contact pressure between the sliding plate and the contact conductor is approximately 70 N, and the contact pressure will change periodically and randomly due to the pantograph–catenary vibration [9]. In addition, the sliding speed of the sliding plate and the conductor is relatively high, and the current is conducted strongly, generally between 100 A and 1000 A. It must withstand not only the relative sliding friction at a high speed of more than 500 km/h but also the ablation of the off-line arc. The pantograph catenary system has been in the exterior environment, while the working environment is chaotic and diverse, and the process of live friction and wear is complex. It is affected by mechanical, current, and thermal effects as well as working conditions [10,11]. A good contact condition between the two is necessary to ensure the train’s safe, stable operation and the quality of the current collection [12]. The running speed of the electric locomotive is constantly increasing, which puts forward higher requirements for the overall performance of the pantograph slide plate. As the traditional type cannot meet the requirements, it is urgent to develop a new pantograph slide plate material with high strength, high conductivity, wear resistance, and good pantograph–catenary coupling. Domestic and foreign researchers are dedicated to the research of composite materials [13,14,15], especially carbon fiber composite materials [16,17,18,19,20,21]. Carbon fiber occupies an important position in advanced composite reinforcements because of its excellent properties, such as low thermal expansion coefficient, low density, excellent green resistance, high modulus, strength-to-weight ratio, superior electrical conductivity, and resistance to chemical reaction [22,23,24,25,26]. Carbon fiber composite materials have high specific strength, corrosion resistance, wear resistance, and are self-lubricating [27,28]. The material properties can be designed through component control, and it is expected to replace the traditional sliding plate and become a new sliding plate material. Wen and others proposed a simple and effective electrostatic method to spray a small amount of negative charges on the surface of carbon fibers to obtain excellent wettability and improve the mechanical properties of composites [29]. Zhou et al. have developed the Mo_2_C layer that was fabricated throughout the internal surface of a C/C preform in molten salt with ammonium paramolybdate and successfully produced C/C−Cu composite materials with good interfacial bonding, as well as good bending and impact strength, achieving resistivity of the same order of magnitude as pure copper [30]. Cui and coworkers prepared a C/C−Cu composite reinforced by novel carbon fiber 2.5D-braided preforms by pressureless infiltration technology, which has excellent mechanical properties as well as good electrical conductivity [31].

As a sliding plate material, carbon fiber composite should firstly reduce its resistivity to meet the material selection requirements. Copper is a common material for electrical and electronic components and cables, and it has better thermal conductivity and electrical conductivity than ordinary metals [32]. If a three-dimensional conductive network structure can be formed, the conductivity can be greatly improved. In this work, in order to achieve a strong interfacial bond to optimize the performance for a variety of applications [33,34,35,36], the structure of the carbon fiber composite sliding plate was designed creatively to meet the requirements for electrical conductivity as well as good mechanical and wear resistance properties [37]. Firstly, the influence of three forms of copper is analyzed, clearly showing that the addition of copper mesh can significantly reduce the resistivity. Due to the addition of 20 wt% copper mesh, the resistivity is 11.2 μΩ·m, reduced by 91.77%; the wear loss decreased by 13.21%, and the impact strength increased by 14.19%. Therefore, it is determined that layered copper mesh is the ideal structure of resin-based composites. Notably, the resistivity is related only to the quality of the copper mesh. In the case of a certain quality of copper mesh, the reduction of its thickness and the increase of its aperture are conducive to improving the performance. The best copper mesh size is 5#. The contact mode of the sliding plate and bracket is copper mesh with conductive adhesive, with the minimum contact resistance and the slowest temperature rise of 2.27 °C/min. Given the above discussion, this work aims to provide an effective and feasible strategy to complete the structural design of carbon-fiber-reinforced resin matrix composites and to provide effective technical support for the wide application of composite materials [38].

## 2. Material and Methods

### 2.1. Materials

The pantograph slide plate consisted mainly of thermosetting resin, reinforcing material, conductive phase, lubricating phase, and toughening agent. Carbon fiber (T300) was produced by Shanghai Sijute Fiber Co., Ltd. (Shanghai, China). Phenolic resin (40–60 μm) was purchased from Jinan Shengquan Group Co., Ltd. (Jinan, China). Copper powder (99.5%, 74 μm), copper fiber (0.1 mm), and copper mesh (99.9%) were obtained from Tianjin Damao Chemical Reagent Factory (Tianjin, China) and Hebei Anping Huayi Network Industry Co., Ltd. (Hengshui, China), respectively. Flake graphite (Natural flaky texture 99%) was purchased from Qingdao Tianhe Graphite Co., Ltd. (Qingdao, China). Nitrile rubber (N41) was produced by Nanjing Shengdong Chemical Co., Ltd. (Nanjing, China). Deionized water was self-made and used in all experiments. The comprehensive properties of carbon fiber used in the experiment are shown in Table 1.

### 2.2. Preparation of Pantograph Slide Plates

The preparation schematic is shown in Figure 1. The surface of carbon fiber was modified by the liquid phase oxidation method and coupling agent method, and dried after deionized water cleaning. After adding anhydrous ethanol to the raw materials according to the proportion in Table 2 and mixing evenly, the carbon fiber was soaked in the mixed solution and dried at low temperature for standby. Then, the anhydrous ethanol solution of short-cut fiber and a proportional amount of phenolic resin, graphite, etc., was mixed well in a JWW11-SHX-JJ-125W type mixer and dried below 70 °C and set aside. JWW11-SHX-JJ-125W was produced by Beijing Zhongxi Chemical Glass Instrument Co. (Beijing, China). The impregnated carbon fiber and copper mesh were cut to the size of the mold cavity. The dried materials and copper mesh were filled into the mold sequentially according to the design ratio and were pressed under the preset molding process parameters. The last step was to heat treat the pressed sliding plate sample at 180 °C for 2 h. 

Preparation of samples with different forms of copper fillings: The experimental variable was to select three different forms of copper powder, copper fiber, and copper mesh to determine the best form of copper to add. The specimens of the composite sliding plate without Cu, with Cu powder, Cu fiber, and Cu mesh were labeled as Non−Cu, Cu−P, Cu−F, and Cu−M, respectively. Preparation of samples with different sizes of copper mesh filling: The copper network size in Table 3 is the experimental variable, and six size models of copper mesh named n# were selected to aid in choosing the best copper mesh size. The first part was to study the effect of the size of the copper mesh on the performance of the sliding plate by laying different types of copper mesh and well-mixed fillers into the mold sequentially according to the same mass ratio, and the specimen number was designated as S_n_. The second part was to lay different types of copper mesh into the mold sequentially according to the same number of layers and well-mixed fillers; thus, 8 layers of copper mesh were laid. The size of the copper mesh on the performance of the sliding plate was studied, and the specimen number was designated as T_n_. The test specimens were prepared with different connections to the bracket. Four types of stainless steel mesh, copper mesh, conductive adhesive, and copper mesh with conductive adhesive were used at the bottom of the pantograph to realize the contact between the pantograph slide plate and the bracket and fastened with screws. 

### 2.3. Characterization

The surface morphology and impact fracture surface of samples were tested by scanning electron microscopy (SEM, SU-70, Hitachi, Tokyo, Japan). The Archimedes drainage method was used to test the density of carbon fibers. The resistivity of carbon fiber was measured by four terminal methods, and the average value was taken after ten measurements. Carbon fiber specimens were prepared for testing according to the national standard GB/T3362-2005, and the tensile properties were measured using a Shenzhen SANS CMT4204 (Shenzhen Sansi Material Testing Co., Shenzhen, China) universal testing machine at a speed of 2 mm/min. The temperature of the slide was measured with power-on time to determine the best way to contact the slide with the bracket.

The impact strength was tested on the XJJ-50 (Chengde Jinjian Testing Instruments Co., Ltd, Chengde, China) impact testing machine. The impact toughness of the material was measured by the pendulum experiment. In strict accordance with JB/T8762, ISO179, GB/T2611, and GB/T1043 standards, the sample size was 15 mm × 20 mm × 120 mm, and impact velocity was 2.9 m/s. The sample was unnotched, and the calculation formula of impact strength *δ* (kJ/m^2^) is given as follows:(1)$$\delta =\frac{E}{b\times d}\times 10^{3}$$
where *b* and *d* represent the width (mm) and thickness (mm) of the measured specimen, respectively. *E* is the impact energy absorbed by the measured sample. The average value of five samples tested in each group was the impact strength of the sample.

The resistivity of carbon fiber and samples was measured by an H2MR high-precision resistance tester. Since the copper net was laid inside the sample, which was anisotropic, the resistance of the sample cannot be measured by point contact. A sample holder as shown in Figure 2 was designed to measure the resistance. During the test, the sample was set into the sample rack, the copper sheet and insulating sheet were placed therein, and the bolts were tightened to ensure reliable contact between the copper sheet and the surface of the sample. Voltage and current were applied to the copper sheet at both ends, and the resistance of the sample was measured. The test time was 1 min, counted every 4 s, and the average value of the above test resistance was computed. The resistivity of the sample was calculated according to the following formula: (2)$$\rho_{r}=\frac{US}{IL_{U}}=R\frac{S}{L}$$
where *ρ_r_* is the resistivity of the test sample (μΩ·m). *U* is the voltage value (V) applied to the sample, and *I* is the current value (A). *R* is the measured specimen resistance (Ω), and *S* and *L* represent the sectional area (m^2^) and length (m) of the test specimen, respectively.

The wear amount was tested by MM-200 friction and wear tester, as shown in Figure 3. The opposite grinding material was a copper wheel, with a diameter of 40 mm, a revolution of 400 r/min, a load of 200 N, and wear of 1 h. The torque value was recorded every 15 min. The friction coefficient *f* and abrasion loss *W* of the sliding plate type were calculated according to the following formulas:(3)$$f=\frac{l}{F\times r}$$
(4)$$W=\frac{m_{0}-m_{1}}{F\times 2\pi rnt}$$
where *f* is the friction coefficient, *l* is the average value of torque, *F* is the loaded load, and *r* is the radius of the copper wheel. *m*_0_ is the mass before wear, and *m*_1_ is the mass after wear; *n* is the number of revolutions of the copper wheel, and *t* is the time.

## 3. Results and Discussion

### 3.1. Effect of Copper Morphology on the Performance of the Sliding Plate

For studying the influence of copper morphology on the impact performance, the impact strength of sliding plate materials prepared by different copper morphologies was measured, as shown in Figure 4a. The sliding plate sample without copper was used as a contrast. It can be seen from the figures that the different forms of copper had significantly different effects on impact properties. The addition of copper powder did not change the impact strength much, increasing it by only 1.28%. The addition of copper fiber and copper mesh could improve the impact strength to varying degrees, and the effect of copper fiber was more significant, relatively increasing by 38.53%. The reason for this result is that copper is a ductile material and the addition of copper fibers absorbed the impact energy and delayed or stopped the expansion of cracks at fracture, causing a significant increase in the impact strength of the material. Although the addition of copper mesh could improve the impact strength to a certain extent, due to the poor compatibility between the copper mesh and the matrix material, the distribution of the copper mesh in the matrix was not uniform. In addition, the thickness of the copper mesh was larger than the diameter of the copper fiber, so it could not be embedded into the matrix to form a relatively tight bond. This can explain why the impact strength increased only by 14.19%, which is much less significant than the effect of copper fibers.

In order to further analyze the influence of the different forms of copper on the properties, the impact fracture of the filled sample was measured and is shown in Figure 4c. The morphologies in Figure 4c_1_ and Figure 4c_2_ are similar, and the addition of copper powder had little effect on the impact strength and cross-section. This is consistent with the previous results and analysis. Figure 4c_3_ is an impact cross-sectional view of a copper fiber-filled sliding plate. When subjected to impact load, the copper fiber was broken and presented a ductile fracture. The fiber absorbed part of the impact energy in the process of breaking, so the impact strength of the material was improved as a whole. Figure 4c_4_ is the impact cross-sectional view of the sliding plate filled with copper mesh. Under the impact force, the copper mesh also deformed, which contributed to the improvement of impact strength. However, the improvement of impact strength was not as obvious as that of the specimens with the addition of copper fibers due to the larger thickness and size of the copper mesh and the poor wettability with the resin matrix, which made the pores between the copper mesh and the matrix larger and affected the transfer of load from the matrix. Therefore, the observation and analysis of impact fracture morphology are consistent with the previous impact performance analysis, and the above conclusions are confirmed.

To explore the influence of copper morphology on the resistivity of the sliding plate, the resistivity of materials made of copper with different morphology was tested. It can be seen from Figure 4a that the copper morphology was closely related to the resistivity, and the addition of copper components could effectively reduce the resistivity. Although the resistivity of the sample added with copper powder and copper fiber was significantly lower than that without copper fiber, the resistivity was still large, which does not meet the requirements of sliding plate materials for resistivity. The resistivity of the sliding plate sample with copper mesh was the smallest, which was 11.2 μΩ·m. The reason is that the conductive components were mainly copper, graphite, and carbon fiber. The resistivity of copper was approximately 1.75 × 10^−2^ μΩ·m, while the resistivity of graphite and carbon fiber were 6~10 μΩ·m and 19~25 μΩ·m, respectively. The resistivity of copper was several orders of magnitude lower than that of graphite and carbon fiber, and the layered copper network more easily formed a penetrating copper conductive network structure. The copper mesh with less content could reduce the resistivity, and the critical value of forming a conductive network was low. Therefore, the pantograph slide plate with a layered structure had excellent electrical conductivity, which was superior to the sliding plate with the other two forms, and the resistivity of the sample without copper mesh was reduced by 91.77%. Figure 4d shows SEM and EDS diagrams of sliding plate samples filled with copper in the different forms. It can be seen from Figure 4d_1_,d_2_ that the copper powder was evenly distributed inside the sliding plate, and there was less contact between particles, so the resistivity of the sliding plate material was correspondingly high. This is proved by the resistivity results measured in Figure 4a. Figure 4d_3_,d_4_ represent the morphology of the copper-fiber-filled sample, in which the copper was relatively concentrated. Although there was contact between the copper fibers, no continuous three-dimensional conductive network was formed, and the resistivity of the specimens without the added copper mesh was reduced significantly. Only by increasing the content of copper fiber was it possible to form a three-dimensional network structure and effectively reduce the resistivity. From the morphology diagrams of the copper-mesh-filled slide plate observed in Figure 4d_5_,d_6_, it can be seen that the distribution of copper inside the slide plate was not uniform and anisotropic, which allowed it to more easily form a copper conductive network structure for electron circulation. The final material with good electrical conductivity was obtained, which was easy to prepare for the pantograph slide plate. The interpretation of SEM diagrams here is consistent with the change of sample resistivity, which confirms the accuracy and reliability of the analysis.

The effect of copper morphology on the frictional properties is analyzed from Figure 4b. The change of copper morphology had little effect on the friction coefficient, and its value always fluctuated between 0.24 and 0.26. The coefficient of friction of the sliding plate specimens increased slightly with the addition of copper components; with the addition of copper powder, copper fiber, and copper mesh specimens, it increased by 5.38%, 6.45%, and 1.44%, respectively. At the same time, the wear resistance of the sliding plate specimens with the addition of copper powder performed the best, with a 39.3% reduction in wear, followed by copper fiber, with a 22.96% reduction in wear. The wear amount of the sliding plate specimen with the addition of copper mesh was reduced by 0.41 × 10^−5^·N^−1^·m^−1^, which is 13.21% lower compared with the specimen without the addition of copper. On the one hand, the reason for this phenomenon is that the addition of copper increased the adhesive force between the composite material and the copper wheel, which increased the friction coefficient of the sliding plate and the adhesive wear. On the other hand, due to the excellent thermal conductivity of copper, the addition of copper facilitated the transfer of a large amount of frictional heat to the interior of the specimen, thus reducing the temperature of the friction surface and making the material more wear-resistant. Therefore, the addition of copper mesh had a positive effect on the friction performance of the sample. Based on the above analysis, the addition of copper mesh not only had no negative impact on the impact performance and wear resistance of the sliding plate but also improved the comprehensive performance of the sliding plate. Notably, when the mass fraction of copper was 20%, the addition of copper mesh effectively reduced the resistivity of the composite material to 91.77%, which could meet the requirements of the pantograph sliding plate on the conductivity of the composite material. Furthermore, the impact strength of the sliding plate sample with copper mesh increased by 14.19%, and the wear loss decreased by 13.21%. Therefore, the layered structure is the ideal structure for the resin matrix composite pantograph slide. Figure 5 is the cross-sectional view of the pantograph slide plate sample with a layered structure, wherein the white dispersion is the side of the copper mesh. Next, the effect of the size effect of the copper mesh on the performance of the skids is further investigated in the context of the above conclusions.

### 3.2. Influence of Size Effect of Copper Mesh on the Performance of the Sliding Plate

The pantograph slide plate is a current-collecting element for the electric locomotive to obtain power from the conductor of OCS, and it needs to conduct current in the process of use. Under the relative sliding friction of the special friction pair composed of mechanical and electrical coupling, friction heat and Joule heat are constantly generated. Therefore, it is required that the material has excellent electrical and thermal conductivity. As the conductive component of the pantograph slide plate with a layered structure, the size and distribution of copper mesh affect the performance. The wettability of the mixture of copper mesh and phenolic resin is poor, and it easily cracks under the action of external force, thus affecting the product performance. Therefore, it is imperative to analyze the influence of the size effect of copper mesh on the performance of sliding plates. Due to the different thicknesses and apertures of the copper mesh, the number and distribution of the copper mesh in the sliding plate must be different according to the same mass of the copper mesh. Table 4 shows the number of layers of copper mesh laid in the sample under the same mass of copper mesh.

In order to study the influence of the size effect of copper mesh with the same quality on the sliding plate performance, the resistivity, impact strength, and wear amount were measured. To ensure the same quality of the copper mesh, the resistivity of the sliding plate material made of different sizes of copper mesh was measured and is shown in Figure 6a. When the mass of copper mesh was certain, the effect of changing the size of copper mesh on the resistivity of the sliding plate was small, and the change in sample resistivity was not significant. This is because the resistivity of the sliding plate is determined by the resistance of the copper network. It is further explained that the resistance of the copper network is determined by the length and area of the copper network. When the mass and length of the copper mesh are fixed, the area must be equal, so the resistivity of the sliding plate has little relationship with the size of the copper mesh. 

Under the condition of the same copper mesh quality, the impact strength of the sliding plate corresponding to different copper mesh sizes was measured, as seen in Figure 6a. It can be seen from the figure that the impact strength of the sliding plate samples (S_1_, S_2_, S_3_, and S_4_) gradually decreases with the decrease of the copper mesh aperture and the number of layers. In samples S_5_ and S_6_, due to the reduction of the thickness of the copper mesh, the impact strength of the sliding plate material is improved. This is because the existence of the copper mesh cut the matrix, and the compatibility between the copper mesh and the matrix material was poor. Under the premise of the similar thickness of copper mesh, the smaller the aperture of copper mesh, the smaller the proportion of matrix material penetrating between copper meshes. Therefore, with the decrease of the copper mesh aperture, the adhesion between the composite and the copper mesh became worse, and the impact strength showed a downward trend. The thickness of the latter two types of copper mesh was smaller than that of the first four types of copper mesh, which was convenient for the penetration of composite materials, such as fibers in the aperture of the copper mesh, and it reduced the cleavage degree of the matrix. Therefore, the impact strength of S_5_ and S_6_ was higher than that of slide plate samples with similar pore size of copper mesh. At the same time, with the decrease of the pore diameter, the impact strength had the same trend as that of the first four samples. The abrasion loss of sliding plate materials made of different sizes of copper mesh is shown in Figure 6c, where the mass of copper mesh was the same. It can be seen from the figure that the abrasion loss of S_1_, S_2_, S_3_, and S_4_ materials increased with the decrease of the copper mesh aperture. When the thickness of the copper mesh decreased, the wear resistance increased, which was similar to the trend of the impact strength. This indicates that the pore size of the copper mesh became smaller, which cut the matrix and the material could not penetrate the copper mesh to form a unified whole. When subjected to friction, it easily fell from the base, so the wear resistance of the product was poor. When the thickness of the copper mesh decreased and the number of layers increased, the total contact area between the composite and the copper mesh increased, which was conducive to improving the bonding property and obtaining the composite with improved wear resistance.

The same test was conducted to explore the influence of the size effect of copper mesh with the same distribution on the performance of the sliding plate. During the working process of the pantograph slide plate, when the current is transmitted, the excessively concentrated copper mesh will form a contraction resistance, causing a large amount of heat to be generated locally. This will not only affect the working quality of the sliding plate, but also increase the friction and wear, and ultimately shorten the service life of the sliding plate. Therefore, when designing the sliding plate structure, we must try to ensure that the copper mesh is evenly distributed in the sliding plate. Due to the different sizes of copper mesh, the mass of copper mesh with the same distribution must be different. The mass of copper mesh in each sample, when eight layers of copper mesh were evenly laid in the sliding plate material sample, is shown in Table 5.

The impact strength of sliding plate material with the same number of copper mesh layers is shown in Figure 6b, from which one can analyze the impact strength of the copper mesh size effect. From the changing trend of impact strength of the T_1_, T_2_, T_3_, and T_4_ samples, it can be seen that the impact strength increased firstly and then decreased with the decrease of aperture. The reason is that the copper mesh is a soft metal with good toughness. When subjected to impact force, the copper mesh can absorb part of the impact energy by deforming or changing the crack propagation direction. Therefore, the impact strength of the sliding plate can be improved to a certain extent with the increase of the mass of the copper mesh. As the pore size continues to decrease, the bonding between the copper mesh and the rest of the composite decreases, resulting in the composite not being able to penetrate through the mesh. The impact strength of the sliding plate material will be reduced when the cleavage of the copper mesh on the material is dominant. Since the thickness of copper mesh in T_5_ and T_6_ samples was reduced, the impact strength was improved. From the point of view of the increase of the impact strength of the sliding plate, the impact strength of the T_5_ sample was higher than that of the T_2_ sample and slightly lower than that of the T_3_ sample. This experimental result shows that in terms of improving the impact performance of the sliding plate, the reduction of the thickness of the copper mesh had a better effect than the increase of the mass of the copper mesh. 

Figure 6b shows the resistivity of sliding plate material prepared by copper mesh with the same number of layers and different sizes. It can be seen from the figure that the resistivity of the sliding plate decreased gradually with the decrease of the copper mesh aperture and the increase of the mass. This is because the copper mesh affected the resistivity. It can be seen from the previous analysis that the resistivity of the composite material after adding copper mesh was reduced by two orders of magnitude, which confirms the statement that copper mesh is the main conductive component in the sliding plate. Therefore, with the increase of the mass of the copper mesh, the electrical conductivity of the sliding plate material must be improved accordingly. The abrasion loss of the sliding plate with the same number of copper mesh layers is shown in Figure 6d. Based on this, the influence of copper mesh size effect on friction properties is studied. It can be seen from the figure that the abrasion loss of T_1_, T_2_, T_3_, and T_4_ samples showed a trend of first decreasing and then increasing. From the latter two samples, the reduction of the thickness of the copper mesh was beneficial in reducing the wear amount of the composite and improving the wear resistance of the material. The influence of copper mesh on friction performance is determined by two factors. On the one hand, the increase of copper mesh content makes the thermal energy generated by the friction surface conduct to the inside of the sample faster. The friction surface temperature of the material can be prevented from being too high and the wear resistance of the sliding plate can be improved. On the other hand, because copper has a face-centered cubic structure, it easily produces adhesive wear. Therefore, when the content of copper mesh is too high, the wear amount increases with the increase of the content of copper mesh.

From the above analysis, it can be concluded that the larger the aperture and the smaller the thickness of the copper mesh, the better the comprehensive performance of the sliding plate under the premise of a certain mass of the copper mesh. The increase of the aperture or the decrease of the thickness of the copper mesh will inevitably increase the number of layers of the copper mesh. The composite materials between the layers are too small to fill the aperture of the copper mesh and cover the whole plane, which will also affect the comprehensive performance. Therefore, it must be ensured that the increase of copper mesh aperture and the decrease of copper mesh thickness are within a proper range. According to the distribution of composite materials during the preparation of the samples, the number of layers of copper mesh should not exceed sixteen. Compared with the performance of pantograph slide plates made of 1# and 5# copper grids, the resistivity had little relation to the size of copper grids. The impact strength and wear resistance of 1# copper mesh and 5# copper mesh were similar, but due to the large aperture and poor strength of 1# copper mesh, it easily deformed during cutting and preparation. Therefore, it is determined that the size of 5# copper mesh is the best copper mesh size for the pantograph slide plate, the aperture is 3.16 mm × 6.3 mm, and thickness is 0.48 mm.

### 3.3. Determining the Contact Mode between the Sliding Plate and Bracket

The contact mode between the pantograph slide plate and bracket directly affects the current transmission quality. Poor contact will increase the contact resistance between the sliding plate and the bracket, resulting in power loss. At the same time, part of the electric energy will be converted into heat energy in the transmission process, causing overheating at the contact point. Finally, the temperature of the sliding plate will be too high, which accelerates the wear of the sliding plate, and then affects the performance of the sliding plate material. Therefore, it is absolutely necessary to determine the best connection mode between the sliding plate and the bracket. Stainless steel mesh, copper mesh, conductive adhesive, and copper mesh with conductive adhesive are used at the bottom of the pantograph slide plate to realize contact with the aluminum bracket. The relationship between contact mode and temperature rise of the pantograph slide plate is shown in Figure 7. It can be seen from the figure that the sliding plate connected with stainless steel mesh had the highest temperature rise and the fastest temperature rise speed. The temperature rise of the sliding plate contacted by copper mesh with conductive adhesive was the minimum, and the temperature was 92 °C after 30 min of heating, which is only 54.46% of that of the stainless steel sample. At the same time, its temperature rise curve had the smallest variation and the smallest temperature rise rate of 2.27 °C/min. Under the same test conditions, the temperature rise rates of stainless steel mesh, copper mesh, and conductive adhesive were 4.83 °C/min, 3.83 °C/min and 2.5 °C/min, respectively. Therefore, the contact resistance between the sliding plate and the bracket connected by copper network with conductive adhesive was the minimum.

The temperature of the sliding plate connected with conductive adhesive was significantly lower than that of the sliding plate connected with metal mesh, which indicates that adding conductive adhesive on the contact surface was conducive to the conduction between the sliding plate and the aluminum bracket. When no conductive adhesive was added, the contact between the sliding plate and the bracket occurred through the screws which connected the sliding plate, metal mesh, and aluminum bracket as a whole. The contact area of this connection method was smaller, and with the extension of the energization time, the temperature of the contact point was increased by a large current, which easily caused oxidation and deformation of the metal. Due to the different expansion coefficients of materials, the increase of temperature further reduced the contact area, resulting in the increase of contact resistance and high temperature, which aggravated the adverse consequences of sliding plate wear. The addition of conductive adhesive converted the contact between the sliding plate, and the bracket into surface contact increased the contact area and reduced the contact resistance, so the connection temperature was lower than that of the first two. The copper mesh with conductive adhesive can be used to control the thickness of conductive adhesive so as to better transmit current; thus, it is an ideal connection method.

## 4. Conclusions

In short, the optimization of the mechanical and electrical conductivity properties is accomplished by comparing the morphology of copper and the effect of the size effect of the copper mesh on the performance of the sliding plate, as well as the effect of the contact mode between the sliding plate and the bracket on its temperature rise. The resistivity of the specimen with the addition of 20 wt% copper mesh is 11.2 μΩ·m, the impact strength is increased by 14.19%, and the wear is reduced by 13.21%; hence, it is determined that the laminarly laid copper mesh structure is the ideal structure for the petals. Importantly, the resistivity is related only to the mass of the copper mesh but not to the size or distribution of the copper mesh; specifically, the increase in the mass of the copper mesh and the decrease in the thickness are beneficial in improving the impact performance and wear resistance. Remarkably, the best type of copper mesh is 5#; additionally, the sliding plate and bracket are connected by copper mesh with conductive adhesive, which has the slowest heating rate of 2.27 °C/min and the smallest resistance. The next step is to further explore the raw material selection and design of carbon-fiber-reinforced resin-based pantograph slide plates based on the current research conclusions so as to determine the best formula. Therefore, this research work explores the composite laminate structure and provides an effective solution to optimize the performance of the sliding plate. 

## Figures and Tables

**Figure 1 materials-15-07876-f001:**
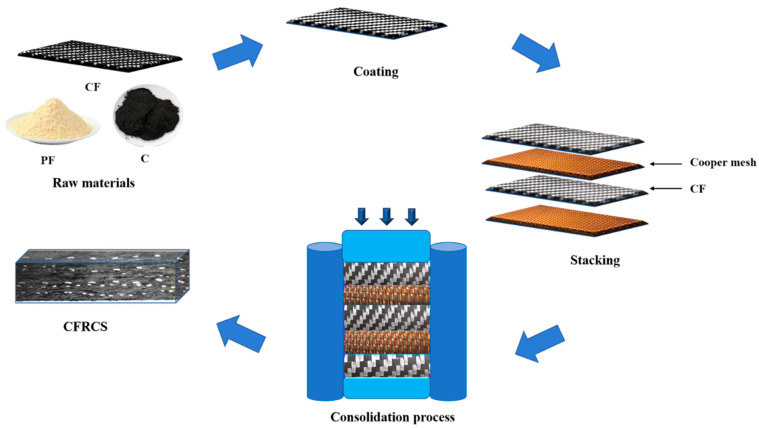
Schematic diagram of pantograph slide plate preparation.

**Figure 2 materials-15-07876-f002:**
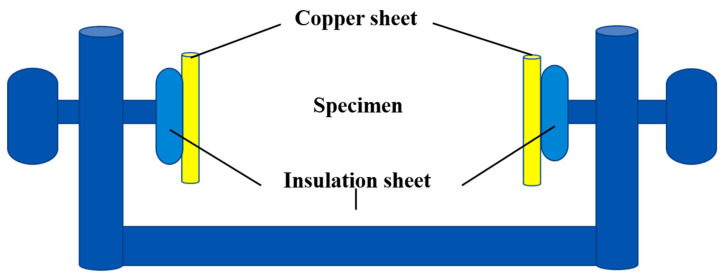
Schematic diagram of resistivity test.

**Figure 3 materials-15-07876-f003:**
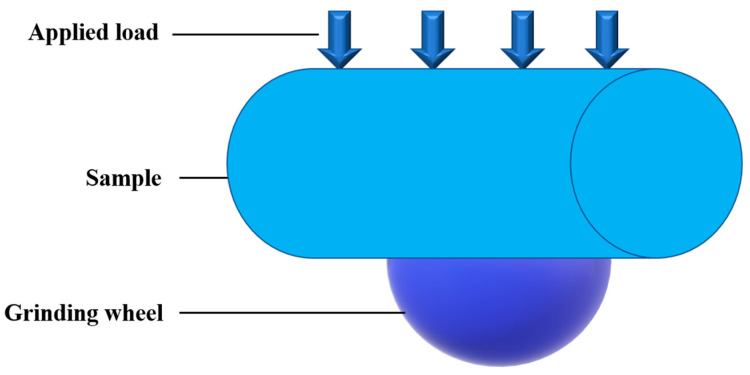
Schematic diagram of wear testing equipment.

**Figure 4 materials-15-07876-f004:**
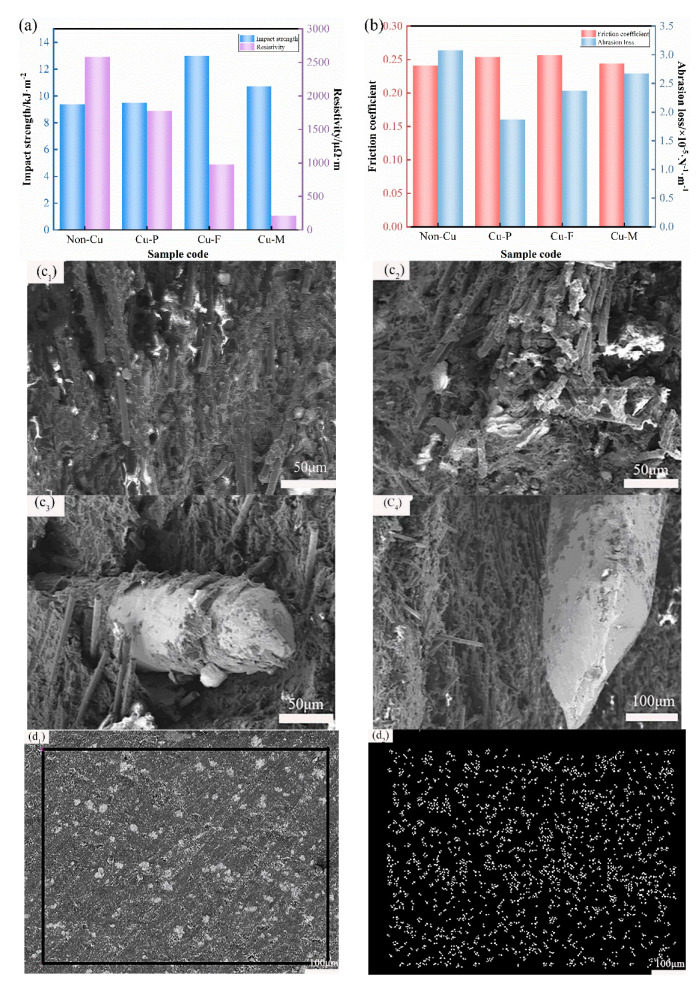
(**a**) Impact strength and resistivity and (**b**) friction coefficient and abrasion loss of Cu−filled slide specimens with different morphologies. Impact fracture morphology of the Cu−M−filled slide specimens of (**c_1_**) Non−Cu, (**c_2_**) Cu−P, (**c_3_**) Cu−F, and (**c_4_**) Cu−M. SEM and EDS images of (**d_1_**,**d_2_**) Cu−P; (**d_3_**,**d_4_**) Cu−F; and (**d_5_**,**d_6_**) Cu−M−filled slide specimens.

**Figure 5 materials-15-07876-f005:**
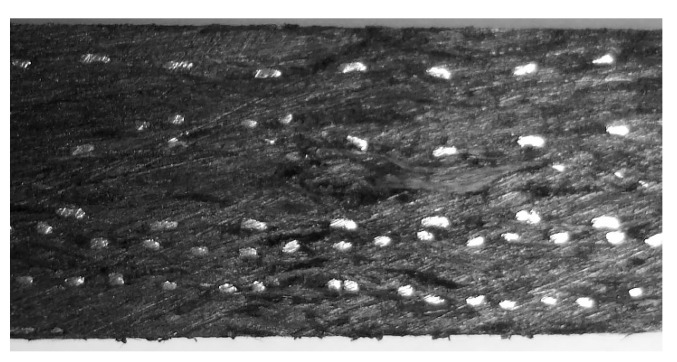
Sectional drawing of layer structure pantograph contact strip.

**Figure 6 materials-15-07876-f006:**
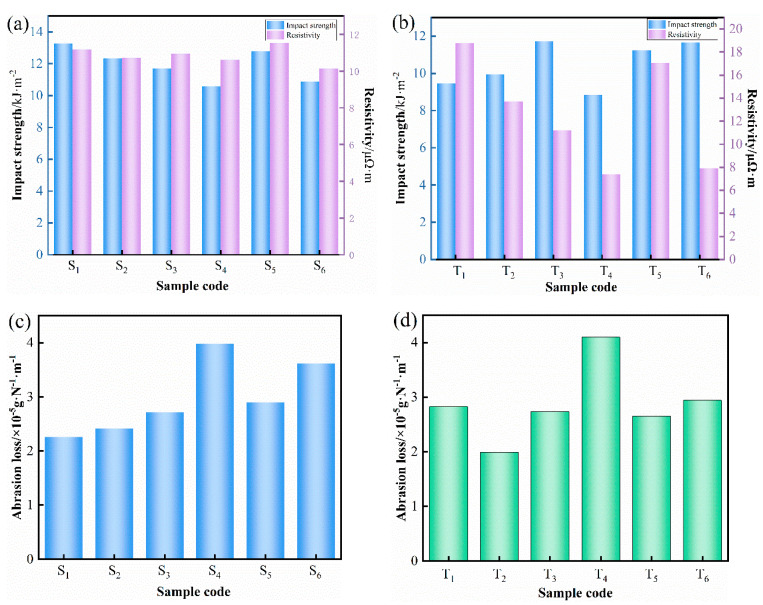
(**a**) Impact strength and resistivity and (**c**) abrasion loss of sliding plate of copper mesh with the same mass and different sizes. (**b**) Impact strength and resistivity and (**d**) abrasion loss of sliding plates with the same size but different copper meshes.

**Figure 7 materials-15-07876-f007:**
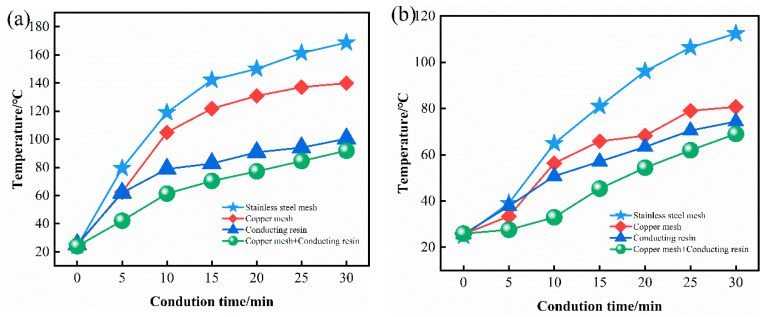
Temperature-time relationship of sliding plate under different contact modes. (**a**) Maximum temperature and (**b**) minimum temperature.

**Table 1 materials-15-07876-t001:** Comprehensive properties of carbon fiber.

Performance	Value
Tensile strength/GPa	3.964
Volume density/g·m^−1^	1.76
Linear density/g·m^−1^	0.2006
Resistivity/μΩ·m	18.2
Elasticity modulus/GPa	246.4
Elongation/%	1.9

**Table 2 materials-15-07876-t002:** The ratio of raw materials.

Number	Carbon Fiber (wt%)	Phenolic Resin (wt%)	Nitrile Rubber (wt%)	Flake Graphite (wt%)	Addition
Non−Cu	21	39	11	29	No additives
Cu−P	21	39	11	29	20 wt% Copper powder
Cu−F	21	39	11	29	20 wt% Copper fiber
Cu−M	21	39	11	29	20 wt% Copper mesh

**Table 3 materials-15-07876-t003:** Copper mesh parameters.

Model Number	Thickness/mm	Aperture/mm·mm	Unit Mass/×10^−4^ g·mm^−2^
1#	0.65	5.47 × 10.29	2.24
2#	0.66	4.43 × 8.11	2.98
3#	0.67	3.28 × 6.36	3.93
4#	0.61	2.30 × 4.06	5.52
5#	0.48	3.16 × 6.30	2.59
6#	0.48	1.96 × 3.07	4.79

**Table 4 materials-15-07876-t004:** Distribution of layers of copper mesh in the sliding plate.

Sample	Number of Layers
S_1_	15
S_2_	11
S_3_	8
S_4_	6
S_5_	13
S_6_	7

**Table 5 materials-15-07876-t005:** Mass of copper mesh in sliding plate materials.

Sample	Mass/g
T_1_	3.37
T_2_	4.58
T_3_	6.02
T_4_	8.14
T_5_	3.86
T_6_	7.45

## Data Availability

Not applicable.

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
