# Peer review of "Laminate Design of Carbon-Fiber-Reinforced Resin Matrix Composites for Optimized Mechanical Properties and Electrical Conductivity"

_materials, 2022, doi:10.3390/ma15227876_

Round 1

Reviewer 1 Report

Please see the pdf file below.

Author Response

Dear Reviewer,

Thank you very much for your letter and the comments concerning our manuscript entitled “Laminate design of carbon fiber reinforced resin matrix composites for optimized mechanical properties and electrical conductivity” Manuscript ID: materials-1964281. We are grateful for the insightful comments and valuable suggestions which have enabled us to improve our work. The manuscript has been carefully revised addressing the issues you raised (marked with red font color in revised manuscript). We hope it is satisfied.

We would like to say thank you for allowing us to resubmit this revised copy of the manuscript. If you need any other information, please do not hesitate to contact us by email.

Thanks for your consideration and best regards.

Yours sincerely,

Hongxue Tan

wangyanxiang0820@163.com

Reviewer 2 Report

The subject investigated in the current study is interesting. Additionally, the  proposed outcomes are considerable.
In my opinion, the literature review should be extended. The novelty and motivation should be explained with more details. Moreover, the results depicted in Figs. 6 and 7 should be described with more details.

Author Response

(The authors gave the same response as above.)

Reviewer 3 Report

The paper will be ready for publication after major revision.

Kindly, revise the paper based on the attached pdf file. 

Author Response

(The authors gave the same response as above.)

Round 2

Reviewer 3 Report

Accept.